

# A 30-year ocean front datasets based on deep learning from 1993 to 2023 for Northwest Pacific ocean

Yuan Niu[1], Xuefeng Zhang[1], Dianjun Zhang[1]

[1]School of Marine Science and Technology, Tianjin University, Tianjin 300072, China

*Correspondence to*: Dianjun Zhang (zhangdianjun@tju.edu.cn)

**Abstract.** Ocean fronts are critical interfaces between different water masses, profoundly influencing atmosphere–ocean interactions, weather systems, marine ecosystems, and climate regulation. Accurate and long-term observations of ocean fronts are essential for advancing studies in meteorology, oceanography, and climate science. However, no publicly available, long-term ocean front dataset currently exists, and existing detection methods often rely on time-consuming manual labeling or

traditional algorithms with limited accuracy in complex frontal regions. In this study, we release the first publicly available 30-year ocean front dataset (1993–2023) for the Northwest Pacific, generated by applying a deep learning framework (Mask R-CNN) to daily sea surface temperature (SST) fields, with manually annotated samples for model training. The dataset provides pixel-level frontal boundaries along with associated attributes, including position, intensity, and width, stored in NetCDF-4 format at 1/12° spatial and daily temporal resolution. Accuracy evaluation shows a mean average precision (mAP)

exceeding 0.90, with smaller errors in front width and intensity compared with traditional gradient-based methods, while capturing more small-scale features. The dataset offers three main contributions: (1) Filling the critical gap of a standardized, long-term ocean front product; (2) Serving as a ready-to-use training resource for deep learning models, greatly reducing the need for manual labeling; and (3) Providing benchmark samples for validation and intercomparison of other ocean front detection products. This dataset supports robust investigations of seasonal-to-interannual frontal variability and provides a

valuable foundation for applications in meteorology, ecosystem management and climate change research.

## 1 Introduction

Ocean front refers to a narrow transitional zone between two or more types of water bodies with significantly different properties, which is a jumping zone of marine environmental parameters and can be described by the horizontal gradient of seawater temperature (Chen 2009; Wang et al. 2015). As an important intersection between the atmosphere and the ocean,

ocean front holds a significant position in the field of Earth science (Azevedo et al. 2021; Lachkar et al. 2011). These fronts are the places where different air masses (usually cold and warm humid air) interact, not only having profound impacts on meteorology and climate, but also playing key roles in ecology, resource management, and climate regulation (Chen 2009; O"Neill et al. 2003; Strass et al. 2002). From extreme weather events to changes in marine ecosystems, to the stability of the global climate system, the importance of ocean fronts cannot be ignored in multiple fields (Belkin et al. 2009; Yu et al. 2019).

Ocean front recognition is of crucial importance for meteorological and climate research (Chronis 2021). By accurately identifying and tracking ocean fronts, it is possible to better understand the dynamics of climate and weather systems, thereby providing early warning for extreme weather events (Saldías et al. 2021). The identification of ocean fronts can help us better





understand the Earth's climate and ecosystem, providing support for global climate change research (Ruiz 2019).

The data sources for ocean front extraction include SeaWiFS water color data (Belkin and O"Reilly 2009), MODIS sea
surface temperature (SST) (Nieto et al. 2012), and AVHRR SST remote sensing image data (Shaw and Vennell 2001). At
present, the main methods for extracting ocean fronts based on remote sensing data include statistical histogram method
(Belkin and Cornillon 2003), gradient method (Breaker et al. 2005; Kostianoy et al. 2004), Canny algorithm (A et al. 2015;
Nieto et al. 2012; Ping et al. 2014; Rameshkumar et al. 2012), wavelet analysis (Park and Il-Seok 2003), the mathematical
morphology method (Cun-Jin et al. 2008; Lea and Lybanon 1993; Qingling and Jianyu 2010), the entropy theory algorithm
(Gen-Yun et al. 2012; Park and Il-Seok 2003; Shimada et al. 2005) and gravity model based methods (Sun et al. 2007). The
main methods for calculating temperature gradients are Gradient method and Sobel gradient algorithm. However, these
algorithms may not effectively distinguish between genuine ocean fronts and other image features or artifacts. They can
produce false positives or miss actual fronts, compromising the accuracy and reliability of the detection results. Traditional
methods often lack the adaptability to handle varying oceanographic conditions and environmental factors. They may struggle
to detect ocean fronts in regions with weak or ambiguous front edges, where the information is less pronounced and subject to
fluctuations. This limitation restricts their application in dynamically changing marine environments. Since 1970s, significant
progress has been made in the detection and research for ocean fronts. The gradient algorithm is a commonly used ocean front
detection method, such as the Sobel operator, Prewitt operator, Laplacian operator, and other gradient operators (Davis, 1975).
However, these operators are too sensitive to noise and have poor performance in detecting small edges. Ping et al. (2014)
proposed a dual I value ocean front recognition method based on the gradient I value method. The Sobel operator was used to
calculate the gradient map of the SST image, and the dual I value interval was determined using the gradient histogram method,
ultimately completing the recognition of the ocean front (Ping et al. 2014). Traditional gradient threshold methods rely on
setting a threshold value to identify sea fronts manually (Chuan-Yu and Fan 2009; Fan et al. 2009; Oram et al. 2008). However,
the selection of this threshold is subjective and lacks a standardized criterion. Different researchers or studies may choose
different threshold values, leading to inconsistency and variability in the detected ocean fronts. Traditional methods based on
gradient thresholds often struggle to accurately detect complex and diverse ocean fronts. These methods may overlook subtle
variations in the gradient values or fail to capture the intricate patterns and transitions associated with complex fronts. This
limitation hampers the ability to comprehensively study and understand the dynamics of ocean fronts. In summary, traditional
methods for extracting ocean fronts suffer from limitations such as subjective threshold selection, inadequate handling of
complex fronts, dependency on edge detection algorithms, and limited adaptability to changing conditions. Overcoming these
limitations is essential for achieving accurate and comprehensive detection of ocean fronts.

With the continuous deepening of deep learning research, convolutional neural networks (CNN) and R-CNN have achieved
great success in various scenarios such as image recognition, speech recognition, and mouth recognition (Chen et al. 2020;
Markus et al. 2019; Yang et al. 2018). On this basis, the Mask R-CNN network achieved pixel level instance segmentation of
images (He et al. 2017). Lima et al. (2017) proposed a fine-tuning neural network for ocean front detection based on previous
research to address the practical situation where deep networks such as AlexNet, Caffe Net, GoogLeNet, and VGGNet are
prone to overfitting under limited training data. Sun et al. (2019) proposed a multi-scale detection framework for ocean front
detection and fine-grained positions  Li et al. (2019) proposed an ocean front recognition network based on CNN to address
the weak edge of ocean fronts. In order to detect more precise ocean fronts, the network fused the convolutional features at
each stage and used the IOU loss function and binary cross entropy loss function to fix model mistakes. Xie et al (2021) used
LSENet to detect and locate multiple ocean fronts in color SST gradient maps, achieving an ocean front recognition
breakthrough with a mDSC higher than 90% .  Li et al. proposed a deep learning model with U-Net architecture that was
designed to detect and locate significant frontal zones in grayscale sea surface temperature images and successively developed
a bidirectional edge detection network (BEDNet)(Li et al. 2020) and weak edge identification network (WEIN) (Li et al. 2022).
Niu et al. designed a multi-scale model Simple and Quick Net (SQNet) for identifying the position of ocean fronts based on
their characteristics (Niu et al. 2023). Felt et al. proposed machine learning (ML) models to detect temperature and chlorophyll
ocean fronts from unprocessed and radiometrically uncorrected satellite imagery by transfer learning from the existing models
for edge detection (Felt et al. 2023).





Table 1. Overview of the existing classic research on ocean front extraction based on deep learning approaches

| Author | Experimental area | Network model | Result accuracy | Advantages | Limitations |
|---|---|---|---|---|---|
| Lima et al. (2017) | Small regions | CNN | 88% | Method involves CNNs and transfer learning via finetuning | Low image resolution |
| Sun et al. (2019) | Small regions | AlexNet | 90% | Six scanning scales | The experimental area is too small |
| Li et al. (2022) | 30-34°N, 132-140°E | U-Net | 90% | Small time cost | The recognition effect of complex sea areas needs to be verified |
| Niu et al. (2023) | The coast of China and the Gulf of Mexico | SQNet | 90% | Based on a multi-scale | The research area is small |
| Felt et al (2023) | Coastal regions | CNN | 90% | Improved computational efficiency | Too few samples in the dataset |

With the application of depth learning in the field of image recognition (Nogueira et al. 2016), in view of the shortcomings of traditional ocean front detection methods, the ocean front detection algorithm based on depth learning has become a research hotspot. Sufficient training samples are the basis of target detection based on deep learning. The fusion of ocean front detection and deep learning can build enough training data with high integration cost and scarce data, especially in the weak ocean front area where the information of ocean front edge is changeable and not obvious, which increases the difficulty of data set

construction. Therefore, considering the small amount of data and weak edge of ocean front, this paper proposed a new automatic detection method of ocean front, which applies Mask R-CNN network to ocean front detection, and then realizes high-precision detection of ocean front through multiple iterative training and parameter correction.

## 2 Study area and data

### 2.1 Study area

The study area for this research is spanning a latitude range of 0° to 50°N and a longitude range of 100° to 150°E (Fig.1). The research area includes Bohai Sea, Yellow Sea, East China Sea, South China Sea, and Western Pacific. These waters cover various ecosystems such as coastal plains, deep trenches, islands, and coral reefs. The marine hydrological conditions are diverse, covering temperate, subtropical, and tropical waters. The changes in ocean temperature, salinity, and ocean currents have significant impacts on marine ecology and climate. The convergence of ocean currents such as the East China Sea Warm

Current, Kuroshio, and Philippine Current in this region has a significant impact on marine ecology and climate change. And it has abundant marine resources, including fishery resources, oil and natural gas reserves, mineral resources, as well as renewable energy such as wind and tidal energy. By specifically examining this region, the research aims to gain insights into the dynamics of ocean fronts and their characteristics in this area. Understanding the behavior and distribution of ocean fronts in the South China Sea is crucial for various applications, including marine ecology, fisheries management, and weather

prediction. The chosen geographic extent provides a representative and comprehensive view of the oceanic features and processes occurring in this dynamic and economically important region.

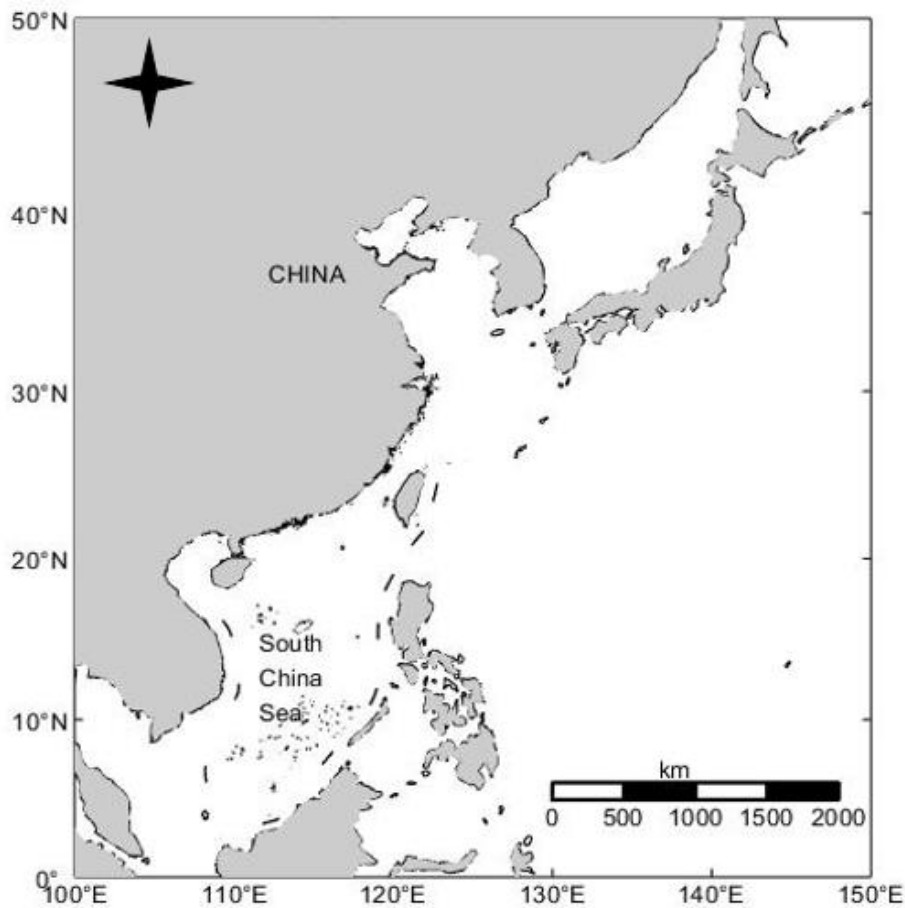

Figure 1. Location of the study area

## 2.2 Data sources

The worldwide ocean eddy-resolving (1/12° horizontal resolution, 50 vertical levels) reanalysis encompassing the altimetry (1993 forward) is provided by the Copernicus Marine Environment Monitoring Service (CMEMS) and is available as the GLORYS12V1 product. It is based largely on the current real-time global forecasting CMEMS system. The model component is the Nucleus for European Modelling of the Ocean (NEMO) platform driven at surface by European Centre for Medium-Range Weather Forecasts (ECMWF) ERA-Interim then ERA5 reanalyses for recent years. Observations are assimilated by

means of a reduced-order Kalman filter. Along track altimeter data, Sea Surface Temperature, Sea Ice Concentration and in-situ Temperature and Salinity vertical Profiles are jointly assimilated. Moreover, a 3D-VAR scheme provides a correction for the slowly-evolving large-scale biases in temperature and salinity. This product includes daily and monthly mean files for temperature, salinity, currents, sea level, mixed layer depth and ice parameters from the top to the bottom. The global ocean output files are displayed on a standard regular grid at 1/12° (approximately 9 km) and 50 standard levels. The data used in

this article is the daily average sea surface temperature dataset, covering a period of 30 years from 1993 to 2023. The unit is Celsius, the time resolution is day, and the spatial resolution is 1/12°. The specific parameter information of the data is shown in Table 2 (https://data.marine.copernicus.eu/product/).





Table 2. Data parameter description

| Information | Details |
|---|---|
| Full Name | Global Ocean Physics Reanalysis |
| Product ID | GLOBAL_MULTIYEAR_PHY_001_030 |
| Source | Numerical models |
| Spatial extent | Global Ocean |
| Spatial resolution | $0.083° \times 0.083°$ |
| Temporal extent | 1 Jan 1993 to 31 Dec 2023 |
| Temporal resolution | Daily, Monthly |
| Elevation levels | 50 |
| Processing level | Level 4 |
| Variables | Cell thickness |
| | Sea water potential temperature (T) |
| | Sea water potential temperature at sea floor (T) |
| Feature type | Grid |
| Blue markets | Polar environment monitoring |
| | Policy & governance |
| | Science & innovation |
| | Extremes, hazards & safety |
| | Coastal services |
| | Natural resources & energy |
| | Trade & marine navigation |
| Projection | WGS 84 (EPSG:4326) |
| Data assimilation | In-Situ TS Profiles |
| | SST |
| Update frequency | Annually |
| Format | NetCDF-4 |
| Originating centre | Mercator Ocean Internationa |

## 3 Methods

### 3.1 Gradient calculation method

In this paper, the gradient method is used to identify and extract the front. The principle of the gradient method is that there is a relatively high gradient near the ocean front. The pixel gradient is calculated and the pixel higher than a certain threshold is selected to extract the ocean front. The gradient method is simple in principle and fast in calculation, so it is widely used in the detection of ocean fronts.

There is a sharp saltation of seawater parameters in the ocean front, and the gradient of seawater parameters can describe the severity of the change. Therefore, it is necessary to calculate the SST gradient, so as to convert the temperature data under



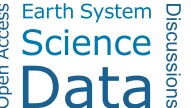

the corresponding longitude and latitude coordinates into gradient data to better characterize the ocean front. Gradient calculations are performed in 3×3 neighborhoods of each data point. The expression of the Gradient calculation is

$$G = \sqrt{D_x^2 + D^2} \tag{1}$$

$$D_x = \frac{T(i, j + 1)T - (i, j - 1)}{2\Delta X} \tag{2}$$


$$D_Y = \frac{T(i+1,j) - T(i,1-j)}{2\Delta T} \tag{3}$$

Where G is the SST gradient; Δ X、 Δ Y is the pixel size in the X and Y directions, respectively. Dx and Dy are the gradient

size in the horizontal and vertical directions, respectively, and i and j are the pixel positions in the image, respectively.

### 3.2 Data label

Labelme software was used to generate the ocean front labels First and foremost, data from remote sensing satellite images
that show ocean fronts must be gathered, which are from various regions, times of year, and types of houses. Then Labelme's marking tool (typically the Polygon tool) was used to mark ocean fronts. Lastly, the label data were saved as a JSON file. 5000 ocean front label datasets were produced through repeating the above-menthioned procedures.

### 3.3 Construction of Mask R-CNN model

Mask R-CNN extends Faster R-CNN by adding a branch parallel to the existing target detection frame to predict the target
mask. Mask R-CNN has three outputs: a class label, a bounding-box offset and the target mask. The difference between the target mask and the class - box output is that it needs a more refined extraction of the target's spatial layout. The network architecture diagram of Mask R-CNN is shown in Fig.2. The architecture takes a square 224 × 224 pixel RGB image as input and produces a distribution over the ImageNet object classes, which is composed of five convolutional layers, three pooling layers, two fc layers, and finally a classifier layer. The success of this architecture is based on several factors, such as
availability of large data sets, more computing power, and availability of GPUs. It also depends on the implementation of additional techniques, such as dropout, data augmentation to prevent overfitting, and rectified linear units to accelerate the training phase.

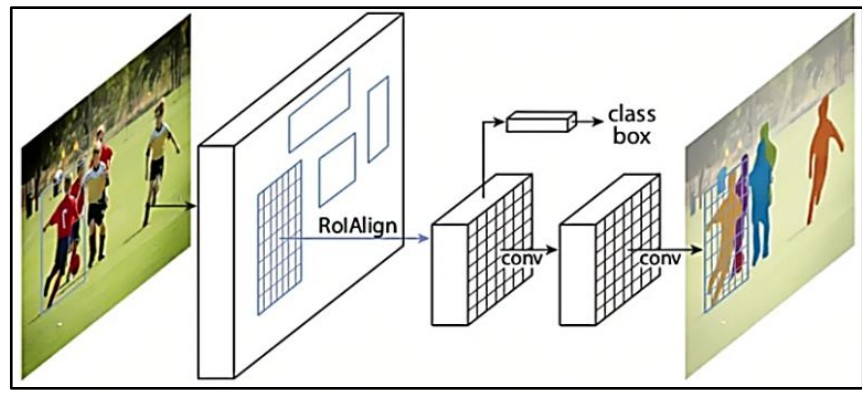

Figure 2. Mask R-CNN network architecture (Gkioxari et al. 2017)


Residual neural network (Res-Net) and feature pyramid network (FPN) are the key networks used to extract features in Mask R-CNN. Res-Net is a residual learning framework to reduce the burden of network training. FPN extracts region of interest (ROI) features from different levels of features according to the size of the features. RPN (region potential network) region generation network is used to generate candidate regions. ROI Align collects image features and candidate region features as the input of the subsequent full connection layer, and then determines the target category.

Applying deep learning methods to ocean front recognition is a challenging task, because fronts have significant visual similarities and are indistinguishable on color and shape. Ocean fronts are regions where there is a sharp transition in oceanic properties such as temperature, salinity and density. These fronts are critical for understanding the dynamics of the ocean and the global climate system. However, detecting and characterizing these fronts is a challenging task due to their complex and dynamic nature. In particular, the visual similarities among different fronts can make it difficult to distinguish them based on

color and shape alone. Deep learning methods offer a promising approach to overcome these challenges. By leveraging large datasets of oceanographic data, including satellite imagery and *in situ* measurements, deep learning models can learn to identify patterns and features that are characteristic of different types of ocean fronts. These models can then be used to classify and characterize fronts with high accuracy and efficiency. To develop effective deep learning models for ocean front recognition, it is essential to carefully curate and preprocess the training data to ensure that it is representative of the range of oceanographic

conditions and front types that may be encountered in the real world. Additionally, the choice of neural network architecture and training parameters have a significant impact on the performance of the model, careful tuning and evaluation are required to ensure optimal results.

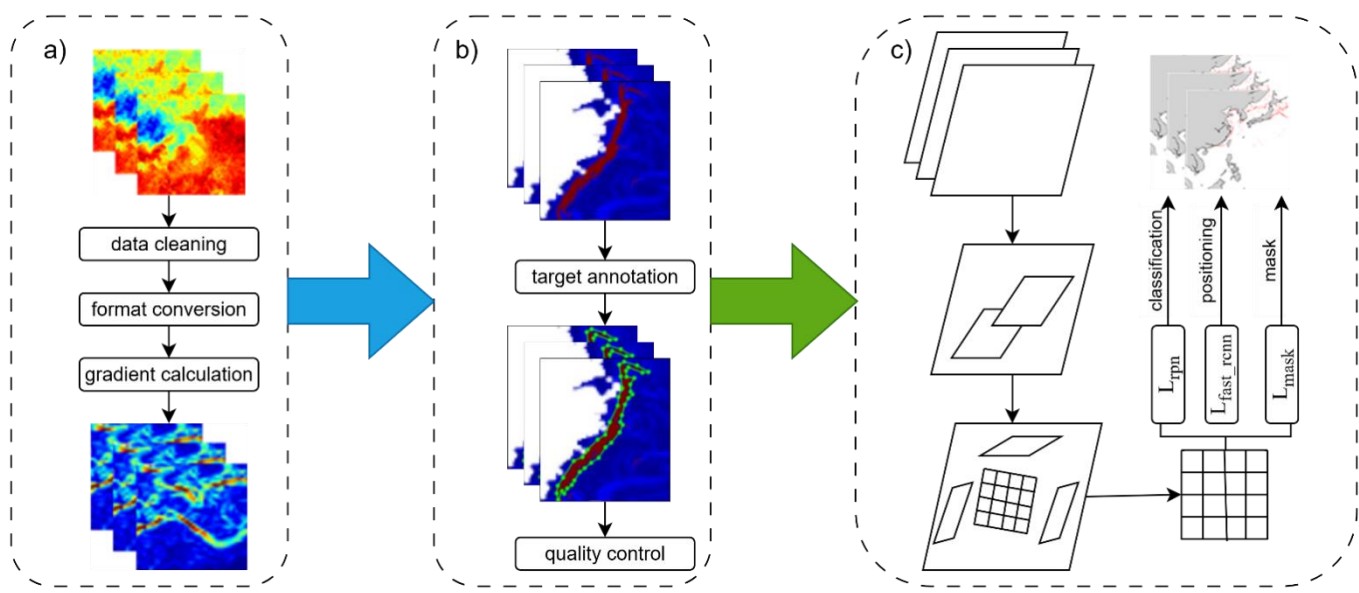


Figure 3. Network architecture diagram for identifying ocean fronts using Mask R-CNN

The loss of Mask R-CNN is the addition of the loss on the Mask branch on top of Faster R-CNN, namely:

$$\text{Loss} = L_{rpn} + L_{fast\_rcnn} + L_{mask} \tag{4}$$

$L_{rpn}$ generates objects by proposing potential bounding box regions in the image. $L_{fast\_rcnn}$ network extracts the proposed

region from RPN and performs region classification and bounding box regression. $L_{mask}$ loss measures the accuracy of the



predicted mask by comparing it with the actual mask. By combining these three losses, Mask R-CNN's overall loss function guides the network to simultaneously perform accurate area recommendations, object classification, bounding box regression and instance segmentation. The network learns to balance these different objectives during training to improve its performance in tasks such as object detection and instance segmentation.

Loss function is usually associated with optimization problems as a learning criterion, that is, to solve and evaluate the model by minimizing the loss function. According to the above task description, the loss function of RPN network training consists of two parts: classification loss and position regression loss.

$$L(\{p_i\}, \{t_i\}) = \frac{1}{N_{cls}} \sum_i L_{cls}(p_i, p_i^*) + \lambda \frac{1}{N_{reg}} \sum_i p_i^* L_{reg}(t_i, t_i^*) \tag{5}$$

$L(\{p_i\}, \{t_i\})$ refers to the entire loss function, representing the total loss of the model. $\frac{1}{N_{cls}}$ is a scalar representing the

normalization factor of the classification loss term, where $N_{cls}$ denotes the number of categories in the classification task. $\sum_i L_{cls}(p_i, p_i^*)$ represents the sum of the classification loss terms, usually used to measure the performance of the model in classification tasks. The classification loss involving each sample $i$, where $p_i$ is the predicted probability distribution of the model and $p_i^*$ is the probability distribution of the actual label. $L_{cls}$ can usually be a cross entropy loss or other classification loss function. $\lambda$ is a non negative constant used to balance the classification loss term and the regression loss term. $\frac{1}{N_{reg}}$ is the

normalization factor for the regression loss term, where $N_{reg}$ represents the number of samples. $\sum_i p_i^* L_{reg}(t_i, t_i^*)$ represents the sum of regression loss terms. The regression loss for each sample $i$ is involved, where $t_i$ is the predicted regression value of the model and $t_i^*$ is the actual regression target value.

In the training process, the gradient image of the SST data on the target date was selected to detect and extract the ocean front. Fig.4 provides an overview of the experimental workflow, illustrating the sequential steps involved in detecting and

evaluating ocean fronts using the Mask R-CNN model. This flow chart helps to visualize the process and highlights the key stages in the experiment. The process can be summarized as follows:

1) SST image: The experiment started with obtaining the SST image data for the study area.

2) Pretreatment: Pre-processing steps for SST image data were conducted to enhance the quality and remove any noise or artifacts. This step is crucial to ensure accurate detection and analysis of ocean fronts.

3) Gradient calculation: The gradient of the SST image was calculated to identify areas of rapid temperature change, which are indicative of ocean fronts. The gradient represents the spatial variation of temperature across the image.

4) Gradient image: The calculated gradient values were used to generate a gradient image, where higher gradient values correspond to stronger temperature gradients and potential ocean fronts. This image provides a visual representation of the potential ocean front locations.

5) Model training: The Mask R-CNN model was trained using the gradient image as input. The model learned to identify and classify the ocean fronts based on the patterns and features present in the gradient image. This step involved training the model on a large dataset with labeled ocean front samples.

6) Detection results: The trained model was applied to the entire dataset of SST images to detect and localize ocean fronts. The model analyzed each image and identified regions where ocean fronts were present.

7) Result evaluation: The detection results were evaluated to assess the performance and accuracy of the Mask R-CNN model in detecting ocean fronts. Various metrics, such as precision, recall, and F1 score, were calculated to measure the model's effectiveness in correctly identifying and locating ocean fronts.





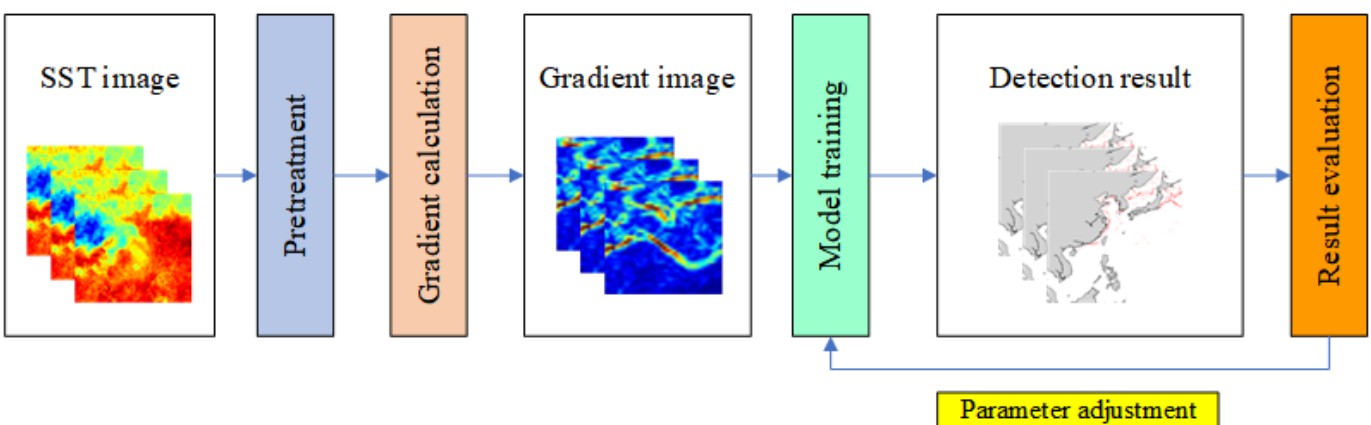

Figure 4. Flow chart of experiment

## 3.4 Mathematical testing methods

Mean average precision (mAP) was used as the evaluation metric. Firstly, the accuracy of the border can be represented by IoU (Intersection over Union), and the general threshold is set to 0.5, which means that if IoU ≥ 0.5, the detection is considered correct. Then, using recall as the horizontal axis and accuracy as the vertical axis, the P-R curve can be obtained. Ideally, both accuracy and recall can achieve results that are infinitely close to 1 at the same time. Therefore, it is hoped that the area covered under the P-R curve is infinitely close to 1. The area below P-R is called the average accuracy of detecting ocean fronts, known as AP (Average Precision). The average of multiple APs is mAP. The definitions of IoU, accuracy, recall, and mean accuracy are as follows:

$$loU = \frac{\text{Area of Overlap}}{\text{Area of Union}} \tag{6}$$

$$P = \frac{TP}{TP+FP} \tag{7}$$

$$R = \frac{TP}{TP+FN} \tag{8}$$

$$P = \int_0^1 P(R)\, dR \tag{9}$$

In the equation, Area of Overlap refers to the intersection of two prediction boxes, Area of Union is the union of two prediction boxes. TP denotes the number of ocean fronts recognized by the model as ocean fronts, FP refers to the number of ocean fronts recognized by other objects, and FN refers to the number of ocean fronts recognized as other objects.



Figure 5. Gradient image of Ocean Front Time Series in January 2020





## 4 Results and discussion

### 4.1 Gradient calculation results

To create a gradient image, the gradient of the original temperature data was computed using formulas (1) through (3). Fig.5 displays the gradient image of the Ocean Front Time Series in January 2023. The gradient image represents the spatial distribution of ocean fronts, indicating areas of sharp temperature gradients within the specified time period. The colors in the image represent different gradient intensities, with warmer colors indicating stronger gradients. The gradient image provides valuable insights into the spatial patterns and variability of ocean fronts during the specified time frame. It allows for a visual

identification of regions with pronounced frontal activity, which is essential for understanding ocean dynamics and processes.

### 4.2 Image marking results

Sufficient training samples are the basis of ocean front detection based on deep learning. As a small scale and weak edge target detection object on SST images, it is difficult to build sufficient and effective training data sets, especially in the weak ocean front area where the edge information of the ocean front is changeable and not obvious. In order to solve this problem, this

paper collected SST images of the regions prone to global ocean fronts, and carried out effective expansion and feature enhancement processing. At the same time, the gradient image was obtained by calculating the gradient from the SST data. Then, each ocean front is labeled to form a set of labeled datasets (Fig.6).

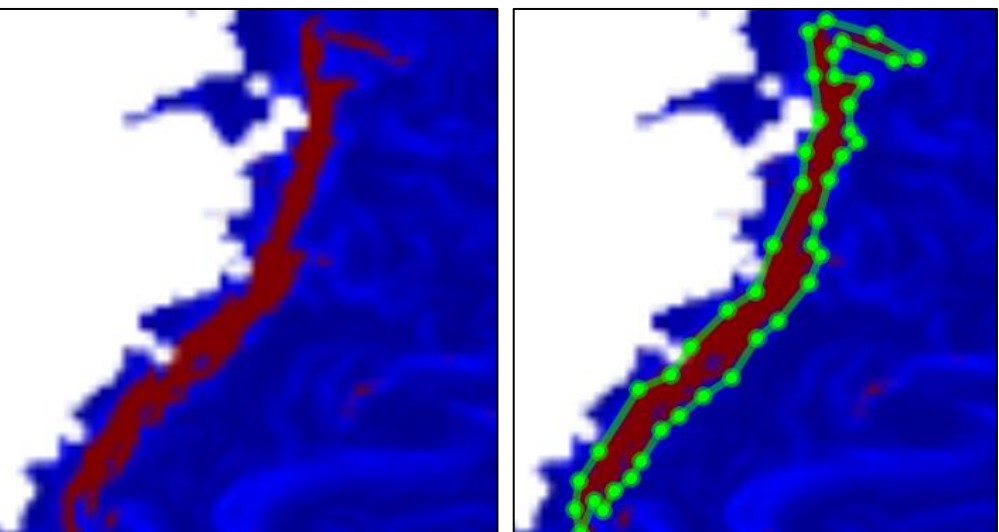

Figure 6. The image before marking (left panel) and after marking (right panel).

### 4.3 Recognition results of the model


In the field of deep learning, the dataset are divided into three parts: training set, validation set, and testing set. The data in the training set is used to establish the model; The validation set is used to evaluate the predictive performance of each model and choose the model with the best performance; After selecting the model with the best accuracy through the validation set, the performance of the optimal model is evaluated through the test set. It is worth noting that in order to ensure the accuracy of

the evaluation, the test set data needs not participate in the training. The quality of the training dataset is the key to affecting the accuracy of the target detection model. To achieve automatic extraction and learning training of high-level essential features of ocean fronts in temperature images by constructing a multi-level network model, thereby achieving automatic detection and



recognition of ocean fronts. The entire process does not require manual intervention, so in the dataset construction process, multi-source temperature data is used to ensure the diversity of training data and enhance the generalization ability of the model.

Mask R-CNN has excellent feature learning ability, but it requires multiple adjustments of various network parameters to optimize the network and detection results, in order to obtain the best ocean front detection model. The network parameters commonly adjusted in deep learning include Momentum, Learning rate, Batch Size, Weight Attenuation Ratio, Iteration Times, and Training Test Data Ratio. Learning rate is usually set to 0.0002, Weight Attenuation is commonly set to 0.0005, and Batch size is set to 2 due to memory limitations. The network parameter settings with the best detection accuracy are mainly obtained by adjusting the proportion of training test data, learning rate, and iteration times.

In comparative experiments, the proportion of training data gradually decreases while the proportion of test data increases, which aims to evaluate the performance of the model under different data distributions. Under given learning rates, batch sizes, and iteration times, the average accuracy (AP) and recall (AR) of the model varied with different data ratios. This indicates that the proportion of training data has a certain impact on the performance of the model. From the given data, it can be observed that in Experiment 1 and Experiment 2, the average accuracy and recall of the model improved when the training data ratio decreased from 0.8 to 0.75. This may be due to the addition of some test data, which enables the model to better generalize to unseen data. However, in experiments 3, 4, and 5, as the proportion of training data further decreased, the average accuracy and recall of the model gradually decreased. This indicates that the model may not be able to fully learn the features of the data with less training data, leading to performance degradation. In addition, we can also observe that there are certain differences in the average accuracy and recall of the model in different experiments. This may be due to changes in the distribution of data, resulting in the model learning different patterns in different experiments. In summary, the proportion of training data has a certain impact on the performance of the model. In some cases, increasing the proportion of test data can improve the performance of the model, but a low proportion of training data may lead to performance degradation. In order to meet the training and testing requirements, 75% of the samples are used as the training set for model training, and 25% as the test set to evaluate the generalization error of the model after training. By continuously adjusting model parameters during model training, the optimal parameter combination is selected according to the detection effect of ocean front.

Table 3. Comparison of training model results under different training and testing ratios

| serial number | The proportion of training data | The proportion of test data | Learning rate | Batch Size | Number of iterations | AP | AR |
|---|---|---|---|---|---|---|---|
| 1 | 0.8 | 0.2 | 0.0002 | 2 | 1000 | 0.918 | 0.893 |
| 2 | 0.75 | 0.25 | 0.0002 | 2 | 1000 | 0.929 | 0.907 |
| 3 | 0.7 | 0.3 | 0.0002 | 2 | 1000 | 0.921 | 0.898 |
| 4 | 0.65 | 0.35 | 0.0002 | 2 | 1000 | 0.902 | 0.846 |
| 5 | 0.6 | 0.4 | 0.0002 | 2 | 1000 | 0.897 | 0.812 |

In order to verify the accuracy of the model detection results, the model is tested with the test data set, the mean average precision (mAP) of the model is calculated, and the results are compared with the manual labeling results, and the intersection overlap (IoU) value is calculated. In addition, in order to further improve the detection results of the ocean front, the model parameters are constantly revised to improve the mAP and IoU. After 30000 iterations, the training network has the ability to detect ocean fronts. The training loss and recognition accuracy (mAP) of the network with different iterations are shown in Table 4. It can be seen that after 30000 iterations, the training has effectively converged and gradually decreased, and the detection accuracy of the model has also increased to 0.9. According to the provided table data, it can be observed that the training loss and recognition accuracy (mAP) of the network improve with the increase of training iterations. At the beginning, as the training progressed to 5000 iterations, the loss was 0.327 and the recognition accuracy was 0.680. As the number of




iterations increases, the loss gradually decreases, while the recognition accuracy gradually increases. At 10000 iterations, the loss decreased to 0.195 and the recognition accuracy improved to 0.770. As the training continues, the loss and recognition
accuracy continue to improve. At 15000 iterations, the loss decreased to 0.156 and the recognition accuracy reached 0.830. At 20000 iterations, the loss further decreased to 0.133 and the recognition accuracy improved to 0.860. After 25000 iterations, the loss decreased to 0.127 and the recognition accuracy reached 0.880. Finally, after 30000 iterations, the loss was further reduced to 0.118 and the recognition accuracy reached 0.920. Based on these observations, it can be concluded that as the number of training iterations increases, the training loss of the network gradually decreases, while the recognition accuracy
gradually improves.

This indicates that the network gradually learns better feature representation and classification capabilities during the training process, thereby improving its performance in recognition tasks. It is worth noting that although both loss and recognition accuracy have significantly improved during the training process, the improvement speed of loss seems to slow down after 30000 iterations, while recognition accuracy still has further improvement. This may indicate that the network has
approached or reached its optimal performance on this dataset, and further training may only result in minor improvements. In summary, based on the provided data, it can be seen that the loss and recognition accuracy of the training network improve with the increase of training iterations, indicating that the performance of the network in learning tasks is gradually improving.

Table 4. Training network loss and recognition accuracy rate

| Iterations | loss | mAP |
|---|---|---|
| 5000 | 0.327 | 0.680 |
| 10000 | 0.195 | 0.770 |
| 15000 | 0.156 | 0.830 |
| 20000 | 0.133 | 0.860 |
| 25000 | 0.127 | 0.880 |
| 30000 | 0.118 | 0.920 |
| 35000 | 0.120 | 0.903 |

In Fig. 7, the recognition results of the ocean front based on the Mask R-CNN model in January 2023 are shown. Through
the application of deep learning algorithms, the position and shape of ocean fronts were identified and displayed using markers and contours. From the graph, it can be observed that the ocean front recognition results based on deep learning algorithms can display obvious features. The position and shape of the ocean front are clearly visible in the figure, indicating that the algorithm can effectively capture the spatial distribution of the ocean front and accurately distinguish it from the surrounding sea area. Secondly, the ocean front recognition results shown in the figure demonstrated the manifestation of small-scale
information. Ocean fronts typically have complex shapes and variations, including slender strip structures and local eddies. Deep learning algorithms can capture these small-scale features, making the recognition results more refined and continuous.

In addition, the ocean front recognition results based on the Mask R-CNN model show good continuity over a time range. The ocean front shown in the figure exhibits a relatively stable distribution and evolution trend over time, indicating that the recognition algorithm has high stability and reliability, and can effectively track and analyze the change process of the ocean
front.

To further test the accuracy of deep learning methods in identifying ocean fronts, the sea surface temperature data for April 2023 was used and ocean fronts were extracted based on traditional gradient method. Fig.8. shows the results of ocean front extracted by traditional methods and depth learning model. It can be seen that the ocean front extracted by depth learning model has a higher fitting degree compared with traditional methods, and can better reflect the small-scale characteristics of
ocean front. More importantly, the deep learning model is more accurate and saves time and effort to identify the ocean front.





Figure 7. Identification Results of Ocean Front Time Series in January 2023







325        Figure 8. Comparison of Ocean Front Results in April 2023. (Left: Gradient method recognition results; Right: Deep learning recognition results)



Calculate the feature elements of the ocean front, extract the intensity and width of the ocean front at the corresponding longitude and latitude based on the recognition results of the ocean front, and thus achieve intelligent extraction of the position, intensity, and width of the ocean front. According to the comparison results in Figure 8, statistical analysis was conducted on the accuracy indicators of ocean fronts identified by traditional methods and deep learning methods, as shown in Table 5. The Monthly Average Gradient Detection Frontal Quantity indicates the average number of detected ocean fronts using gradient detection methods on a monthly basis. On the other hand, the Monthly Average Number of Deep learning Detection Fronts represents the average number of ocean fronts detected using Deep learning methods. The Detection Quantity Accuracy column shows the accuracy of the detection quantity, which is calculated as the percentage of Deep learning detection fronts compared to gradient detection fronts.

For the intensity of single ocean front, the table displays the Gradient and the Deep learning Detection of Single Ocean Front Intensity. Both are measured in degrees Celsius per kilometer. The difference between the two methods is shown in the Error (℃/km) column, which represents the absolute difference (error) between the gradient detection and Deep learning detection of single ocean front intensities. The reported error is 0.013 ℃/km. Furthermore, the table provides information about the width of single ocean front. The Gradient and the Deep learning Detection of Single Ocean Front Width are measured in kilometers. The Error (km) column represents the absolute difference (error) between the gradient detection and Deep learning detection of single ocean front widths. In this case, the error is reported as 0.155 km.

Overall, table V demonstrates the performance and accuracy of the Deep learning method in detecting ocean front characteristics. The high accuracy of the detection quantity suggests that the Deep learning method successfully identifies a significant number of ocean fronts. The small errors in both intensity and width measurements indicate that the Deep learning method closely aligns with the gradient detection method in capturing the characteristics of ocean fronts. These findings highlight the effectiveness of the Deep learning approach for detecting and analyzing ocean fronts.

Table 5. Precision Indicators of Ocean Front Characteristics

| Statistical indicators | Traditional methods | Deep learning | Error |
|---|---|---|---|
| identifying the number of ocean fronts | 11 | 13 | 2 |
| Single Ocean Front Intensity (℃/km) | 0.112 | 0.125 | 0.013 |
| single ocean front width (km) | 27.124 | 27.279 | 0.155 |

**4.4 Analysis of seasonal changes and spatio-temporal characteristics**

The study on seasonal patterns of ocean fronts is extremely important to better understand the Earth's climate system. It helps to gain a deeper understanding of ocean circulation, its impact on seawater temperature, salinity, and nutrient distribution, which is crucial for ecosystems, fisheries, and marine resource management.

The recognition results for the entire year of 2023 were divided into spring (March, April, and May), summer (June, July, and August), autumn (September, October, and November), and winter (December, January, and February) and seasonally averaged to obtain the spatio-temporal distribution results of ocean fronts in spring, summer, autumn, and winter.

From a seasonal distribution perspective, ocean fronts are the most active and numerous in summer, followed by spring. The number of ocean fronts is relatively small in autumn and winter, and the number of ocean fronts is the lowest in winter. In terms of the number of ocean fronts, summer>spring>autumn>winter. From the perspective of spatial distribution, the activity of ocean fronts is relatively frequent in the waters near land. In the South China Sea, ocean fronts are mainly concentrated between Hainan and the Taiwan Strait, and there are many ocean fronts near the Taiwan Strait. In autumn and winter, the activity of ocean fronts in the South China Sea is not active and the number is relatively small.

Overall, spring and summer are the seasons with more oceanic front activity, especially in coastal areas. The number of fronts in autumn and winter is relatively small and not active.


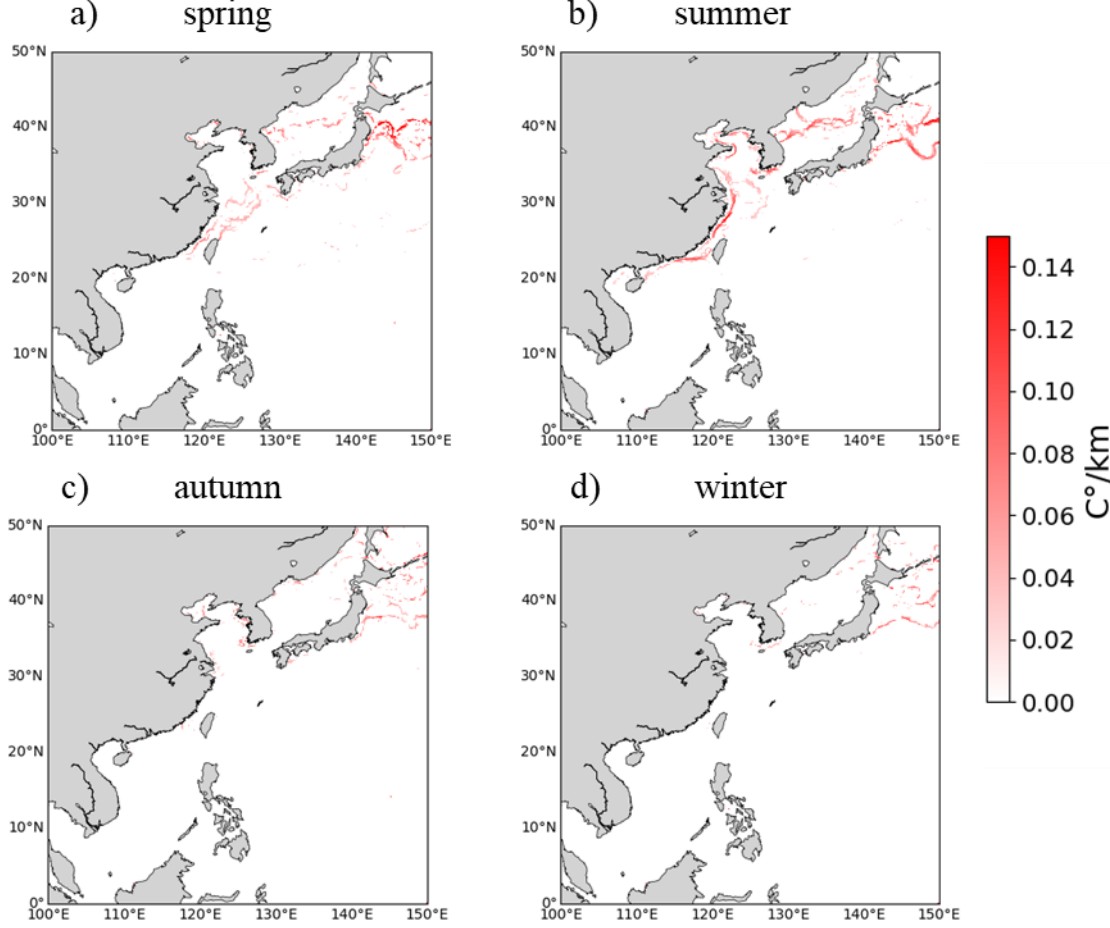

Figure 9. Seasonal spatial-temporal distribution map of ocean fronts in 2023

## 4.5 Discussion

### 4.5.1 Data source errors

The accuracy of ocean front recognition is influenced by data quality. If there is noise, missing values, or inconsistency in the data source, it will cause the model to learn inaccurate features and introduce errors. The reliance on satellite imagery as the primary data source for the proposed method may limit its applicability in certain situations. In such cases, complementary data sources, such as in situ measurements or marine numerical model results, may contribute to enhance the accuracy and robustness of the model.

Meteorological conditions such as wind, clouds, precipitation, and atmospheric pressure may affect the clarity and visibility of satellite images. Adverse weather conditions may make ocean fronts blurry or difficult to recognize in the image. Waves, surges, eddies, and currents also affect the quality of satellite images. These factors may cause image noise or distortion, making the detection of spikes more difficult. The performance of deep learning models depends on the available data. More high-quality data and ground truth data can provide better model training and validation.



### 4.5.2 Model errors

The model errors mainly stems from four aspects: (1) Insufficient model complexity: The selected deep learning model is not
complicated enough to capture complex ocean front patterns, which can lead to errors. (2) Overfitting or underfitting:
overfitting refers to the model performing well on training data but not well on new data, possibly due to the model being too
complex and learning about noise in the training data. Underfitting is due to the model being too simple to capture the
complexity of the data. Unreasonable learning rate settings can also affect the accuracy of recognition. (3) Inaccurate labels:
If the data labels used for training the model are inaccurate and the labeling process is not precise enough, the model will learn
the wrong patterns. (4) Lack of sufficient training data: If the amount of ocean front data available for training is limited, the
model may not be able to fully learn the various features of the front, resulting in errors.

### 4.5.3 The impact of the marine environment

Hydrological conditions: Ocean fronts typically occur at boundaries between different water masses, such as cold and warm
water currents or salinity gradients. The rapid changes in hydrological conditions may affect the shape and position of the front.
Deep learning models should possess the ability to adapt to these changes.

 Seasonal and temporal variation: The position and intensity of ocean fronts may vary significantly in different seasons and
time periods. Deep learning models need to be able to capture these seasonal differences.

 Underwater conditions: Ocean fronts typically exist not only in the ocean surface, but also extend underwater. Underwater
conditions, such as temperature, salinity, and water flow, can also affect the front properties. If relevant underwater data is
available, incorporating it into model training may help improve the  front recognition accuracy.

 In summary, the errors in the recognition results of ocean front deep learning come from the comprehensive influence of
multiple factors such as data quality, data label accuracy, data bias, model complexity, overfitting or underfitting, and training
data volume. In order to improve the accuracy of ocean front recognition, it is necessary to address these factors and
continuously optimize the model and data. Future research should focus on addressing the challenges mentioned above. This
includes developing strategies to obtain more labeled data, improving the model's robustness to environmental factors, and
exploring the potential of integrating different data sources to enhance the accuracy and applicability of the method.

### 5 Data availability

The 30-year ocean front dataset (1993–2023) for the Northwest Pacific is available at https://doi.org/10.5281/zenodo.16921277
(Yuan Niu, 2025).

### 6 Conclusions

An important ocean phenomenon, rapid and accurate detection of ocean front is of great significance to marine ecology, fishery
resources and typhoon path prediction. In view of the scarcity and weak edge characteristics of ocean front data, the data are
expanded in various forms to increase the data set effectively. In this study, an ocean front recognition method based on depth
learning is proposed, which applies Mask R-CNN model to ocean front recognition research to obtain pixel-level ocean front
dataset. The experimental results showed that this method can realize ocean front recognition in an automatic process, and the
recognized front has good independence and integrity. Recognition accuracy reaches over 90%. The  improved Mask R-CNN
model demonstrates superior performance in the detection and extraction of ocean fronts compared to existing methods. The
use of deep learning methods has significantly improved the target recognition ability of ocean fronts, enabling more accurate
detection of fronts, including small-scale fronts. In order to enhance the accuracy of model recognition, cross validation
techniques are used to optimize hyperparameters such as learning rate, batch size, loss function weight, etc. in the algorithm
to achieve better performance. More data augmentation techniques were used to increase the robustness of the model such as



rotation, scaling, flipping, brightness adjustment, etc. Use of 30 years of long-term sea surface temperature data strongly supports a deeper understanding of the seasonal and interannual variations of ocean fronts. By analyzing long-term series data, it is possible to identify the trend of frontal changes, including their seasonal migration and possible climate driving factors. The deep learning method outperforms traditional methods in extracting feature parameters such as ocean front intensity and width. It is possible to extract the characteristic parameters of ocean fronts more comprehensively, and to deeply explore the properties of ocean fronts, such as their scale, intensity, and evolution, which helps to better understand their reaction to the meteorology and climate. Overall, this study demonstrates the effectiveness of the proposed method in detecting and extracting ocean fronts, while also highlighting the need for further research and development to fully unlock its potential for a wide range of applications in oceanography and climatology.

**Author contributions.** XZ and DZ conceived this study. YN and DZ collected the datasets. YN implemented the research and wrote the original draft of the paper. All the authors discussed the results and revised the manuscript.

**Competing interests.** The contact author has declared that none of the authors has any competing interests.

**Acknowledgements.** The authors would like to thank the editors and anonymous reviewers for their valuable comments. The authors thank the Copernicus Marine Environment Monitoring Service (CMEMS) for providing the GLORYS12V1 ocean reanalysis dataset, which is based on the NEMO model and driven by ECMWF ERA-Interim and ERA5 reanalyses.

**Financial support.** This work was supported in part by the Key Research and Development Program, sponsored by the Ministry of Science and Technology (MOST), under Grant 2023YFC3107701 and Grant 2023YFC3107901; in part by the National Natural Science Foundation of China under Grant 42375143.

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
