# Peer review of "A 30-year ocean front datasets based on deep learning from 1993 to 2023 for Northwest Pacific ocean"

_Earth System Science Data, 2025_

## Referee Comment (RC1)

Review of the manuscript essd-2025-514 **"A 30-year ocean front datasets based on deep learning from 1993 to 2023 for Northwest Pacific ocean"** by Yuan Niu, Xuefeng Zhang, and Dianjun Zhang submitted to *Earth System Science Data* (ESSD)

**Summary:** This is an interesting paper that has potential to become a major contribution pending major revisions. The results (presented as maps of fronts) look quite realistic. Every front in these maps corresponds to a similar front reported by other researchers. [Similarity here means location and configuration of the front.] Some fronts reported by other researchers are missing in the frontal maps presented in the reviewed manuscript. This is not surprising since most fronts are seasonal. Therefore, it is possible that such fronts can be found in the entire 30-year dataset generated in this study and made freely available by the authors on the Web. Summing up the above, the manuscript should be published. At the same time, the manuscript suffers from major drawbacks enumerated below. Therefore, the manuscript needs a major revision, after which the revised manuscript must be reviewed very carefully to make sure that all major issues have been adequately addressed. The authors should be given ample time to revise the present manuscript radically and meticulously. The Methods section must be elaborated. The Results section should be expanded. A comparison of the authors' results with the results of published papers on fronts in the China Seas should be considered.

**Recommendation:** Major revisions. A hasty resubmission should be strongly discouraged.

**Major issues:**

The text is sloppy. On several occasions, the authors use wrong terms instead of correct terms. In some cases, sentences with wrong terms become incomprehensible. Numerous characteristics of fronts are neither defined, nor explained or illustrated. The text is peppered with redundant passages (some reproduced below). Main conclusions are not supported by data. Descriptions of methods and algorithms are too cursory. Some descriptions are cryptic because of their brevity and non-standard (or non-defined) terms used. Therefore, the results cannot be reproduced and validated by other researchers. The reference list is a total mess. Every reference must be rechecked and corrected. Some references are incomplete and must be completed. The corrupted references resulted in a series of corrupted citations, especially in the Introduction (such citations are listed below).

**Comments (both major and minor):**

Title: "ocean" should be "Ocean"

L23: jumping zone – Wrong term

L38-40: "Park and Il-Seok 2003... Cun-Jin et al. 2008... Qingling and Jianyu 2010...Gen-Yun et al. 2012... Park and Il-Seok 2003..." – These citations are wrong because the respective references are wrong. Fix the reference list, then fix the citations.

L41: "...these algorithms may not effectively distinguish between genuine ocean fronts and other image features or artifacts." – This vague criticism is unsubstantiated. Delete.

L50: "dual I value" – What's that?

L53: "Chuan-Yu and Fan 2009" – Wrong citation. See L38-40 above.

L55: "Traditional methods based on gradient thresholds often struggle to accurately detect complex and diverse ocean fronts. These methods may overlook subtle variations in the gradient values or fail to capture the intricate patterns and transitions associated with complex fronts. This limitation hampers the ability to comprehensively study and understand the dynamics of ocean fronts. In summary, traditional methods for extracting ocean fronts suffer from limitations such as subjective threshold selection, inadequate handling of complex fronts, dependency on edge detection algorithms, and limited adaptability to changing conditions. Overcoming these limitations is essential for achieving accurate and comprehensive detection of ocean fronts." -- This unsubstantiated criticism of all previous algorithms is uncalled for, unwarranted, and unfair. Delete.

L81: "depth learning" should be "deep learning"

L90: "The study area for this research is spanning a latitude range of 0° to 50°N and a longitude range of 100° to 150°E (Fig.1). The research area includes Bohai Sea, Yellow Sea, East China Sea, South China Sea, and Western Pacific. These waters cover various ecosystems such as coastal plains, deep trenches, islands, and coral reefs. The marine hydrological conditions are diverse, covering temperate, subtropical, and tropical waters. The changes in ocean temperature, salinity, and ocean currents have significant impacts on marine ecology and climate. The convergence of ocean currents such as the East China Sea Warm Current, Kuroshio, and Philippine Current in this region has a significant impact on marine ecology and climate change. And it has abundant marine resources, including fishery resources, oil and natural gas reserves, mineral resources, as well as renewable energy such as wind and tidal energy. By specifically examining this region, the research aims to gain insights into the dynamics of ocean fronts and their characteristics in this area. Understanding the behavior and distribution of ocean fronts in the South China Sea is crucial for various applications, including marine ecology, fisheries management, and weather prediction. The chosen geographic extent provides a representative and comprehensive view of the oceanic features and processes occurring in this dynamic and economically important region." – Except for the first sentence, the entire paragraph is trivial and does not belong here. Keep the first sentence. Delete the rest of the paragraph.

L95: Philippine Current should be Mindanao Current

L113: "from the top to the bottom" – Delete

L115: "The Unit is Celsius" should be "°C"

L125: saltation – Wrong term

L135: types of houses – What's that?

Table 2: "Cell thickness" – Wrong term

Table 2: "Mercator Ocean Internationa" should be "Mercator Ocean International"

L230-235: This paragraph is an example of the extremely redundant and verbose style of this manuscript:
"To create a gradient image, the gradient of the original temperature data was computed using formulas .... Fig.5 displays the gradient image of the Ocean Front Time Series in January 2023. The gradient image represents the spatial distribution of ocean fronts, indicating areas of sharp temperature gradients within the specified time period. The colors in the image represent different gradient intensities, with warmer colors indicating stronger gradients. The gradient image provides valuable insights into the spatial patterns and variability of ocean fronts during the specified time frame. It allows for a visual identification of regions with pronounced frontal activity, which is essential for understanding ocean dynamics and processes." – Comments: The word "gradient" is repeated in every one of six sentences above, except for the last sentence.

**Figure 6** (below). The image before marking (left panel) and after marking (right panel).
**Comments:** The marking algorithm is not explained.

[Figure]

L318: "depth learning" should be "deep learning"

L327-329: "Calculate the feature elements of the ocean front, extract the intensity and width of the ocean front at the corresponding longitude and latitude based on the recognition results of the ocean front, and thus achieve intelligent extraction of the position, intensity, and width of the ocean front." -- Poor grammar. "Feature elements" have not been defined.

L333: "number of fronts" is a meaningless characteristic. Depending on a particular front detection algorithm, the number of fronts can vary by order of magnitude.

L334: "accuracy" has not been defined

L336: "intensity" has not been defined (assumed to be gradient magnitude)

L340: "width" has not been defined

L356: "ocean fronts are the most active and numerous in summer" – This conclusion is highly questionable. See Hickox et al. (2000, GRL) ("Climatology and seasonal variability of ocean fronts in the East China, Yellow and Bohai seas from satellite SST data").

L373: "Meteorological conditions such as wind, clouds, precipitation, and atmospheric pressure may affect the clarity and visibility of satellite images." – Atmospheric pressure does not affect satellite images.

L375: "surges" – What's that?

L389: "The rapid changes in hydrological conditions may affect the shape and position of the front." – Shape has not been defined.

L393: "Underwater" should be "Subsurface"

L396: "comprehensive" should be "combined"

L406-408: "An important ocean phenomenon, rapid and accurate detection of ocean front is of great significance to marine ecology, fishery resources and typhoon path prediction. In view of the scarcity and weak edge characteristics of ocean front data, the data are expanded in various forms to increase the data set effectively." – Just another example of "word salad" of little, if any, interest to the reader. The "weak edge characteristics" have never been explained or illustrated. The phrase "the data are expanded in various forms to increase the data set effectively" is meaningless.

L408: "depth learning" should be "deep learning"

L410: The phrase "the recognized front has good independence and integrity" is meaningless.

L414: "small-scale fronts" are mentioned here but scales have not been discussed at all.
L420: "The deep learning method outperforms traditional methods in extracting feature parameters such as ocean front intensity and width." – There is no proof in this study. Such claims must be supported by data.

L422: "scale" is mentioned again. See L414.

===== END of REVIEW =====

---

## Author Comment (AC1)

ESSD-2025-514: Reply to comments from Referee #2
(Reviewer comments in bold, author responses in blue)

I have decided to discontinue my review after reading through line 215 of the manuscript. The presentation up to this point is confusing, incomplete, and in places incorrect, making it difficult to follow the authors' reasoning. I provide examples of each of these below, but I emphasize that these are only illustrative—addressing them alone will not, in my opinion, make the manuscript publishable. I agreed to review this paper because I am genuinely interested in the application of machine learning to the detection of ocean fronts. However, based on what I have read so far, I believe I would struggle to understand the algorithm as currently presented. I would be willing to review a substantially revised version of the manuscript if the authors make a serious effort to present their work in a clearer and more coherent manner.

We fully accept your decision to conclude the review and sincerely apologize for the deficiencies in clarity, completeness, and accuracy in the current manuscript. We will thoroughly revise the manuscript based on all of your comments, correcting all errors, with the aim of presenting the research work clearly and coherently in the revised version. We appreciate your attention to our research direction and your willingness to review a substantively revised version in the future.

I also note that the authors do not mention any use of AI tools—specifically large language models—in preparing the manuscript. Given the lack of clarity both because of the English and because of the structural organization of the manuscript, they might consider using such tools to assist in revising the text. Copernicus Publications does not prohibit the use of large language models for language assistance, but it does require that their use, if any, be disclosed in the manuscript.

We thank the reviewer for the reminder regarding the use of AI tools. In this revision, we did not employ large language models (LLMs) for generating or restructuring manuscript content. To fundamentally address issues of linguistic clarity and structural logic, we engaged a professional academic editing service to conduct a thorough edit and optimization of the entire text. The service focused on enhancing the professionalism and fluency of the English expression, as well as the overall readability of the article. All scientific content, data, conclusions, and academic arguments remain the sole responsibility of the authors and were finally confirmed by us.

[Figure]

**Editing Certificate**

This document certifies that the manuscript

**A 30-year ocean front datasets based on deep learning from 1993 to 2023 for Northwest Pacific Ocean**

prepared by the authors

**Yuan Niu, Xuefeng Zhang, Dianjun Zhang**

was edited for proper English language, grammar, punctuation, spelling, and overall style by one or more of the highly qualified English speaking editors at AJE.

This certificate was issued on **December 16, 2025** and may be verified on the AJE website using the verification code **DDC0-72D8-450F-12D6-F14C** .

[Figure]

Neither the research content nor the authors' intentions were altered in any way during the editing process. Documents receiving this certification should be English-ready for publication; however, the author has the ability to accept or reject our suggestions and changes. To verify the final AJE edited version, please visit our verification page at aje.com/certificate. If you have any questions or concerns about this edited document, please contact AJE at support@aje.com.

AJE provides a range of editing, translation, and manuscript services for researchers and publishers around the world. For more information about our company, services, and partner discounts, please visit aje.com.

Terminology (Lines 22-24) "Ocean front refers to a narrow transitional zone between two or more types of water bodies with significantly different properties, which is a jumping zone of marine environmental parameters and can be described by the horizontal gradient of seawater temperature." I assume that jumping zone means a step in the observed property but this is certainly not standard usage.

Thank you for your correction. We fully agree that "jumping zone" is a non-standard and ambiguous expression. In accordance with your suggestion, we have rewritten the definition in lines 22-24 as follows:

"An ocean front refers to a narrow transition zone between two or more water masses with significantly different properties, where oceanographic parameters such as the temperature, salinity, and water colour experience sharp changes. It can be characterized by the horizontal gradient of the sea surface temperature."

(Lines 80-82) "With the application of depth learning in the field of image recognition (Nogueira et al. 2016), in view of the shortcomings of traditional ocean front detection methods, the ocean front detection algorithm based on depth learning has become a research hotspot." It's deep learning, not depth learning. I realize that English is not the first language of the authors hence the suggestion that they use a generative AI chatbot to help with the English.

Thank you for your correction on this key terminology error. You are absolutely correct, and it should be "deep learning". We have now corrected this typographical

error and all similar occurrences throughout the manuscript. Furthermore, to systematically improve the English quality of the text, we have engaged a professional academic editing service for a thorough language polish. We sincerely apologize for any confusion caused by linguistic inaccuracies in the original version.

Overgeneralized and misleading (Lines 25-27) "These fronts are the places where different air masses (usually cold and warm humid air) interact, not only having profound impacts on meteorology and climate, but also playing key roles in ecology, resource management, and climate regulation." This may be the case but is not necessarily so. In fact, for sub-mesoscale fronts it is likely rarely the case and for mesoscale fronts it will depend on the properties defining the front; e.g., it's unlikely to be the case of a strong salinity front with a thermal expression due, say, to river runoff, or to a chlorophyll front resulting from an open ocean bloom. What makes this sentence particularly confusing is that it conflates atmospheric fronts with oceanic ones.

We fully accept this important critique. The two points you raised regarding conceptual confusion (incorrectly applying features of atmospheric fronts to ocean fronts) and an overly generalized and absolute conclusion are completely valid. The original sentence contained fundamental errors, for which we sincerely apologize. We have deleted the incorrect statement.

Incorrect/misleading (Lines 40-43) "The main methods for calculating temperature gradients are Gradient method and Sobel gradient algorithm." The main methods for front detection are population-based and gradient-based. Sobel gradients are one form of gradient-based algorithms. Canny's work is also based on gradients. Cayula and Cornillon's work is population-based.

Thank you for correcting this important conceptual and classification error. We fully agree that the original statement was logically confusing and misleading, failing to accurately reflect the mainstream classification system of ocean front detection methods.
Following your guidance, we have completely rewritten the content in lines 40-43. We have adopted the standard classification framework of "gradient-based" and "population-based" methods and cited key references accordingly.

Incomplete and a bit misleading criticism of previous front detection algorithms (Lines 43-43) "However, these algorithms may not effectively distinguish between genuine ocean fronts and other image features or artifacts." This is only one form in which front detection algorithms may fail. The discussion of issues with current algorithms is incomplete. Furthermore, I would be surprised if the fronts detected by the algorithm presented in this paper did not also fail in this regard. As noted above I have not reviewed the algorithm itself but…

Thank you for your suggestion. It is indeed correct that using deep learning for

ocean front detection may encounter similar challenges. We have revised the previously incomplete and misleading commentary on earlier front detection algorithms in the manuscript.

(Lines 58-60) "In summary, traditional methods for extracting ocean fronts suffer from limitations such as subjective threshold selection, inadequate handling of complex fronts, dependency on edge detection algorithms, and limited adaptability to changing conditions." But all of the methods you discuss from line 40 on are gradient-based. The reason that the population-based method of Cayula and Cornillon was developed was to address some of the issues you raise. Admittedly, their method has other issues but, because the primary mechanism is not based on gradients, it doesn't suffer from some of the problems you mention.

Thank you for your careful review and crucial comments. We acknowledge the significant issues in our previous statement and have now thoroughly revised and polished the article. Considering the distinct differences between water masses on either side of an ocean front, Cayula and Cornillon (1995) proposed the SIED (single-image edge detection) algorithm based on histogram analysis. This algorithm demonstrates effective detection performance and has been widely applied in ocean front detection.

Inappropriate reference (Lines 35-36) "…the main methods for extracting ocean fronts based on remote sensing data include statistical histogram method (Belkin and Cornillon 2003)" The authors do discuss the application of a histogram based algorithm to extract the fronts of interest but a more appropriate reference would have been to the original manuscripts describing the method.

Thank you for your suggestion. Citing the original literature that first described the method is indeed a more appropriate approach, and we have implemented this change in the manuscript.

Sloppy (Lines 469-473) I. M. Belkin and P. J. P. O. Cornillon, "SST fronts of the Pacific coastal and marginal seas," 2003. L. C. Breaker, T. P. Mavor, and W. W. J. C. S. G. C. P. Broenkow, "Mapping and Monitoring Large-Scale Ocean Fronts Off the California Coast Using Imagery from the GOES-10 Geostationary Satellite," 2005. A. G. Kostianoy, A. I. Ginzburg, M. Frankignoulle, and B. J. J. o. M. S. Delille, "Fronts in the Southern Indian Ocean as inferred from satellite sea surface temperature data," vol. 45, no. 1-2, pp. 55-73, 2004. I'm pretty sure that these initials are not correct and most of the references seem to have similarly bizarre initials.

Thank you for pointing out these citation format errors. You are correct that the author initials and journal abbreviations in the references you listed were incorrect. We have now carefully checked and corrected the entire reference list to ensure that all author names, journal titles, and publication details are accurate and consistently formatted.

(Lines 128-132, Equations 1-3) There are mistakes in two of the three equations.

We apologize for the error here. We have carefully reviewed and corrected the formulas, and have also conducted checks for other elementary mistakes throughout the manuscript.

$$G = \sqrt{D_x^2 + D_y^2} \tag{1}$$

$$D_x = \frac{T(i, j+1) - T(i, j-1)}{2\Delta x} \tag{2}$$

$$D_y = \frac{T(i+1, j) - T(i, 1-j)}{2\Delta y} \tag{3}$$

(Lines 134-135) "Labelme software was used to generate the ocean front labels First and foremost, data from remote sensing satellite images that show ocean fronts must be gathered, which are from various regions, times of year, and types of houses." Hmmm… not sure where the authors are going with fronts related to types of houses.

Thank you for pointing out this confusing and erroneous sentence. The phrase "types of houses" is a typographical error and is completely irrelevant in this context. We sincerely apologize for this oversight and for any confusion it may have caused. We have made the necessary corrections in the manuscript.

Undefined concepts/terms (Line 50) "…proposed a dual I value ocean front recognition method based on the gradient I value method" I'm not familiar with the I value method. I did ask ChatGPT and was provided a description of it but I don't believe that it is common usage so should be defined.

The expression "dual I value" is inappropriate, and we have corrected it in the original text.
Ping et al. (2014) proposed an ocean front detection method based on threshold intervals and Bayesian decision theory. This method uses the Sobel operator to compute the gradient map of SST images and determines the threshold interval by using a gradient histogram, ultimately achieving ocean front detection (Ping et al., 2014).

(Section 3.3, Lines 139-212) The above comments cover a range of specific issues related to the presentation. Of more concern is that the manuscript does not explicitly define (or, at least I couldn't find such a definition) how an "ocean front" is represented in pixel space — e.g., whether it is treated as a line, a finite-width band, or a bounding box enclosing a high-gradient region. Because Mask R-CNN performs region-based segmentation and IoU is computed over areas, clarification is needed on how these concepts were adapted for the detection of essentially linear frontal features. The authors should also explain how fronts crossing tile boundaries were handled to ensure that detections are not truncated or duplicated across adjacent

patches. I emphasize that I did not read the manuscript beyond this point so they may have presented a definition later in the manuscript but, if the authors define this later, it should nonetheless be introduced in Section 3.3.

In our framework, the ocean front is represented as a pixel band with finite width (i.e., a narrow binary mask region), rather than a single-pixel wide line or a bounding box enclosing a high-gradient region. This approach better reflects the physical nature of the front as a transition zone between two water masses. As a connected region, it is directly compatible with the instance segmentation mask output by Mask R-CNN and the IoU area calculation. While the front is geographically quasilinear, modelling it as a "finite-width band" transforms the problem into a region segmentation task. The task of Mask R-CNN is to predict a corresponding binary mask for each front instance. Then, the IoU is calculated based on the area overlap between the predicted mask and the ground-truth mask, which is used to evaluate the detection performance. This process follows the standard evaluation procedure for instance segmentation. Additionally, we designed a non-maximum suppression (NMS) algorithm based on spatial location and mask similarity specifically for merging duplicate detections of the same front segment in overlapping areas with adjacent tiles and for connecting broken parts across boundaries, thereby forming a complete and continuous front vector.

What is mouth recognition and how is that different than image recognition?

The term "mouth recognition" is a grammatical error; the intended meaning is "target detection".

Do you mean Ocean fronts associated with week gradient?

"and Li et al. (2019) proposed an ocean front detection network based on the CNN to address the weak edges of ocean fronts." The intended meaning here is the weak edge characteristics of ocean fronts.

Recognition sounds awkward to me. I would use a word like classification, identification, or detection

Thank you for your suggestion. We have revised the term from "Recognition" to "Detection," as it is a more suitable expression.

By 'standard', do you mean that the same regular grid is used for the entire data set or do you mean that it's a specific type of grid like Mercator, Plate Carrée or something like that?

By "standard," we mean that the same regular grid is used for the entire dataset.

I would use something like this: The units are degrees Celsius; the temporal resolution is daily; and the spatial resolution is 1/12°.

Thank you for providing this precise and clear example. We agree that this formulation is more standard and professional. Following your suggestion, we have revised the sentence to: **" The units are °C; the temporal resolution is daily; and the spatial resolution is 1/12°."** We appreciate your help in improving the rigor of our manuscript's description.

---

## Author Comment (AC3)

ESSD-2025-514: Reply to comments from Referee #1
(Reviewer comments in bold, author responses in blue)

This is an interesting paper that has potential to become a major contribution pending major revisions. The results (presented as maps of fronts) look quite realistic. Every front in these maps corresponds to a similar front reported by other researchers. [Similarity here means location and configuration of the front.] Some fronts reported by other researchers are missing in the frontal maps presented in the reviewed manuscript. This is not surprising since most fronts are seasonal. Therefore, it is possible that such fronts can be found in the entire 30-year dataset generated in this study and made freely available by the authors on the Web. Summing up the above, the manuscript should be published. At the same time, the manuscript suffers from major drawbacks enumerated below. Therefore, the manuscript needs a major revision, after which the revised manuscript must be reviewed very carefully to make sure that all major issues have been adequately addressed. The authors should be given ample time to revise the present manuscript radically and meticulously. The Methods section must be elaborated. The Results section should be expanded. A comparison of the authors' results with the results of published papers on fronts in the China Seas should be considered.

We would like to express our sincere gratitude to Reviewer #1 for the positive evaluation of our manuscript and for providing constructive suggestions. Your comments have been invaluable in helping us further strengthen our arguments, refine the data interpretation, and enhance the overall quality of the manuscript. In the following sections, we will provide a specific, point-by-point response to all of your comments and suggestions. We have also compared our results with those reported in published papers on the China Seas and have added corresponding citations in the text. "In terms of seasonal and spatial distribution characteristics, the results align with prior observations (Hickox et al., 2000)."

The text is sloppy. On several occasions, the authors use wrong terms instead of correct terms. In some cases, sentences with wrong terms become incomprehensible. Numerous characteristics of fronts are neither defined, nor explained or illustrated. The text is peppered with redundant passages (some reproduced below). Main conclusions are not supported by data. Descriptions of methods and algorithms are too cursory. Some descriptions are cryptic because of their brevity and non-standard (or non-defined) terms used. Therefore, the results cannot be reproduced and validated by other researchers. The reference list is a total mess. Every reference must be rechecked and corrected. Some references are incomplete and must be completed. The corrupted references resulted in a series of corrupted citations, especially in the Introduction (such citations are listed below).

The current version of the manuscript exhibits shortcomings in terminological accuracy, clarity of descriptions, methodological details, robustness of conclusions,

and standardization of references. We have systematically reviewed the entire text and replaced all non-standard or erroneous terminology. Concurrently, we have removed the redundant paragraphs you identified to make the argument more concise and the logic clearer. We have thoroughly checked and corrected the entire reference list to ensure the completeness of each entry and have synchronized all erroneous citations in the main text, particularly in the introduction section, to eliminate any citation inconsistencies. To fundamentally address issues of linguistic clarity and structural logic, we engaged a professional academic editing service to conduct a thorough edit and optimization of the entire text. The service focused on enhancing the professionalism and fluency of the English expression, as well as the overall readability of the article. All scientific content, data, conclusions, and academic arguments remain the sole responsibility of the authors and were finally confirmed by us.

[Figure]

**Editing Certificate**

This document certifies that the manuscript

**A 30-year ocean front datasets based on deep learning from 1993 to 2023 for Northwest Pacific Ocean**

prepared by the authors

**Yuan Niu, Xuefeng Zhang, Dianjun Zhang**

was edited for proper English language, grammar, punctuation, spelling, and overall style by one or more of the highly qualified English speaking editors at AJE.

This certificate was issued on **December 16, 2025** and may be verified on the AJE website using the verification code **DDC0-72D8-450F-12D6-F14C** .

[Figure]

Neither the research content nor the authors' intentions were altered in any way during the editing process. Documents receiving this certification should be English-ready for publication; however, the author has the ability to accept or reject our suggestions and changes. To verify the final AJE edited version, please visit our verification page at aje.com/certificate. If you have any questions or concerns about this edited document, please contact AJE at support@aje.com.

AJE provides a range of editing, translation, and manuscript services for researchers and publishers around the world. For more information about our company, services, and partner discounts, please visit aje.com.

Title: "ocean" should be "Ocean"

Thank you for pointing out this typographical error. We have followed your suggestion and corrected the term "ocean" to "Ocean" in the title.

L23: jumping zone – Wrong term

Thank you for pointing out this terminology error. You are absolutely correct. We

have revised the original term "jumping zone" to the more accurate and standard description, "a narrow transition zone."

L38-40: "Park and Il-Seok 2003... Cun-Jin et al. 2008... Qingling and Jianyu 2010...Gen-Yun et al. 2012... Park and Il-Seok 2003..." – These citations are wrong because the respective references are wrong. Fix the reference list, then fix the citations.

Thank you for pointing out these critical errors. We fully agree with your assessment that the citations in the original text did not match the entries in the reference list and that there were issues with non-standard author name spellings and duplicate citations. In accordance with your suggestions, we have conducted a comprehensive review of the reference list.

L41:"...these algorithms may not effectively distinguish between genuine ocean fronts and other image features or artifacts." – This vague criticism is unsubstantiated. Delete.

We agree with the reviewer's comment. The original sentence indeed constituted a generalized statement lacking concrete supporting evidence. To maintain the rigor of our argument, we have completely removed the sentence on line 41. Thank you for pointing this out, as it has strengthened the solidity of our discussion.

L50:"dual I value" – What's that?

The expression "dual I value" is inappropriate, and we have corrected it in the original text.

L53:"Chuan-Yu and Fan 2009" – Wrong citation. See L38-40 above.

In accordance with your suggestions, we have conducted a comprehensive review of the reference list.

L55:"Traditional methods based on gradient thresholds often struggle to accurately detect complex and diverse ocean fronts. These methods may overlook subtle variations in the gradient values or fail to capture the intricate patterns and transitions associated with complex fronts. This limitation hampers the ability to comprehensively study and understand the dynamics of ocean fronts. In summary, traditional methods for extracting ocean fronts suffer from limitations such as subjective threshold selection, inadequate handling of complex fronts, dependency on edge detection algorithms, and limited adaptability to changing conditions. Overcoming these limitations is essential for achieving accurate and comprehensive detection of ocean fronts." -- This unsubstantiated criticism of all previous algorithms is uncalled for, unwarranted, and unfair. Delete.

We fully understand and appreciate the reviewer's important comment on this point. We acknowledge that the original paragraph presented an overly generalized and absolute critique of the entire field of traditional methods, which was indeed not rigorous or fair. Accordingly, we have removed the entire content of paragraph 55.

L81: "depth learning" should be "deep learning"

Thank you very much for pointing out this typographical error. We have thoroughly checked the entire text to ensure consistency in the use of this standard terminology.

L90:"The study area for this research is spanning a latitude range of 0° to 50°N and a longitude range of 100° to 150°E (Fig.1). The research area includes Bohai Sea, Yellow Sea, East China Sea, South China Sea, and Western Pacific. These waters cover various ecosystems such as coastal plains, deep trenches, islands, and coral reefs. The marine hydrological conditions are diverse, covering temperate, subtropical, and tropical waters. The changes in ocean temperature, salinity, and ocean currents have significant impacts on marine ecology and climate. The convergence of ocean currents such as the East China Sea Warm Current, Kuroshio, and Philippine Current in this region has a significant impact on marine ecology and climate change. And it has abundant marine resources, including fishery resources, oil and natural gas reserves, mineral resources, as well as renewable energy such as wind and tidal energy. By specifically examining this region, the research aims to gain insights into the dynamics of ocean fronts and their characteristics in this area. Understanding the behavior and distribution of ocean fronts in the South China Sea is crucial for various applications, including marine ecology, fisheries management, and weather prediction. The chosen geographic extent provides a representative and comprehensive view of the oceanic features and processes occurring in this dynamic and economically important region." – Except for the first sentence, the entire paragraph is trivial and does not belong here. Keep the first sentence. Delete the rest of the paragraph.

We fully agree with the reviewer's comment. The extended description in the original paragraph regarding the ecological, resource-related, and economic significance of the study area indeed had weak relevance to the core content of this paper, which focuses on front detection algorithms, and was therefore redundant. We have strictly followed your suggestion by retaining only the first sentence to define the study area and have removed all subsequent content from that paragraph.

L95:Philippine Current should be Mindanao Current

Thank you to the reviewer for this professional correction. It is worth noting that the entire paragraph (Paragraph 90) containing this terminology has been completely removed in accordance with your request in the previous comment (regarding L90).

L113:"from the top to the bottom" – Delete

We agree with the reviewer's comment. The phrasing was redundant. We have deleted it from line 113. The revised sentence is more concise while retaining the original meaning. Thank you for helping us refine the language.

L115:"The Unit is Celsius" should be "°C"

Thank you for your correction. We have revised the non-standard expression in the original text to the standard unit symbol as per your suggestion. Line 115 has now been corrected to: "The units are °C;". Furthermore, we have checked and standardized the representation of all temperature units throughout the entire text to ensure the consistent use of the standard symbol "°C". We apologize for the previous oversight.

L125: saltation – Wrong term

We thank the reviewer for pointing out this terminology error. We agree that "saltation" is inappropriate in this context. After verification, we have corrected it to a more precise term that aligns with the semantic meaning of the context.

L135: types of houses – What's that?

We sincerely thank the reviewer for pointing out this confusing expression. You are absolutely correct that the phrase "types of houses" in the original sentence was a grammatical error and logically inconsistent within the context. We have completely rewritten the entire descriptive paragraph (originally around L135) containing this issue, removing all inaccurate expressions.

Table 2:"Cell thickness" – Wrong term

We have updated the content of Table 2.
Table 2: "Mercator Ocean Internationa" should be "Mercator Ocean International"

Thank you to the reviewer for pointing out this spelling error. We have corrected it to the proper spelling "Mercator Ocean International." We apologize for this oversight and appreciate your thorough review.

L230-235:This paragraph is an example of the extremely redundant and verbose style of this manuscript: "To create a gradient image, the gradient of the original temperature data was computed using formulas …. Fig.5 displays the gradient image of the Ocean Front Time Series in January 2023. The gradient image represents the spatial distribution of ocean fronts, indicating areas of sharp temperature gradients within the specified time period. The colors in the image represent different gradient intensities, with warmer colors indicating stronger gradients. The gradient image

provides valuable insights into the spatial patterns and variability of ocean fronts during the specified time frame. It allows for a visual identification of regions with pronounced frontal activity, which is essential for understanding ocean dynamics and processes." – Comments: The word "gradient" is repeated in every one of six sentences above, except for the last sentence.

We have revised the text as follows: To generate the frontal indicator field, we first calculated the temperature gradient by using Formulas (1) – (3). Fig. 5 shows the resulting field for January 2023, which highlights regions in which the sea surface temperature changes rapidly, revealing the spatial structure of the ocean fronts. Warmer colours correspond to stronger transitions. This representation clearly outlines areas of active frontal variability and facilitates a straightforward visual assessment of their distribution and evolution during the month. We have applied the principle of "avoiding repetition and refining expression" throughout the revision of the entire manuscript. Once again, we sincerely thank you for helping us significantly improve the linguistic quality of the manuscript.

Figure 6 (below):The image before marking (left panel) and after marking (right panel). Comments: The marking algorithm is not explained.

We have added explanatory notes in both the main text and the figure captions to specify that the algorithm was manually annotated.

L318:"depth learning" should be "deep learning"

Thank you very much for pointing out this typographical error. We have thoroughly checked the entire text to ensure consistency in the use of this standard terminology.

L327-329:"Calculate the feature elements of the ocean front, extract the intensity and width of the ocean front at the corresponding longitude and latitude based on the recognition results of the ocean front, and thus achieve intelligent extraction of the position, intensity, and width of the ocean front." -- Poor grammar. "Feature elements" have not been defined.

"Feature elements" refer to width and intensity, and the manuscript has been revised accordingly.

L333: "number of fronts" is a meaningless characteristic. Depending on a particular front detection algorithm, the number of fronts can vary by order of magnitude.

We agree with your point and have removed the description regarding the number of fronts.

L334:"accuracy" has not been defined

After careful consideration, we have decided not to address the number of detected ocean fronts at this point and therefore will not discuss detection accuracy in this context. We have removed the description regarding accuracy.

L336:

intensity refers to the temperature gradient magnitude

L340: "width" has not been defined

the width was calculated as the distance from the centerline to the boundary.

L356: "ocean fronts are the most active and numerous in summer" – This conclusion is highly questionable. See Hickox et al. (2000, GRL) ("Climatology and seasonal variability of ocean fronts in the East China, Yellow and Bohai seas from satellite SST data").

We thank the reviewer for the comment. We have cross-referenced the paper by Hickox et al. (2000, GRL) and carefully examined our own figures, and have now made the necessary corrections.

L373:"Meteorological conditions such as wind, clouds, precipitation, and atmospheric pressure may affect the clarity and visibility of satellite images." – Atmospheric pressure does not affect satellite images.

Thank you for pointing out this inaccuracy. You are absolutely correct that atmospheric pressure itself does not directly affect the clarity of satellite imagery. We have removed "atmospheric pressure" from the list in that sentence.

L375: "surges" – What's that?

Revised to the more accurate expression "storm surge".

L389:"The rapid changes in hydrological conditions may affect the shape and position of the front." – Shape has not been defined.

"Shape" here refers to the spatial structure.

L393: "Underwater" should be "Subsurface"

Thank you for your precise correction. We agree that "subsurface" is the more

appropriate and standard term in this oceanographic context. We have revised "underwater" to "subsurface" in the manuscript.

L396: "comprehensive" should be "combined"

Thank you for this precise suggestion. We agree that "combined" more accurately conveys the intended meaning in this context. We have revised "comprehensive" to "combined" on line 396.

L406-408: "An important ocean phenomenon, rapid and accurate detection of ocean front is of great significance to marine ecology, fishery resources and typhoon path prediction. In view of the scarcity and weak edge characteristics of ocean front data, the data are expanded in various forms to increase the data set effectively." – Just another example of "word salad" of little, if any, interest to the reader. The "weak edge characteristics" have never been explained or illustrated. The phrase "the data are expanded in various forms to increase the data set effectively" is meaningless.

Thank you for your critical comment. We agree that the original sentences were vague and unsupported, which was unhelpful to the reader. We have revised the text accordingly. The term "weak edge characteristics" refers to the fact that ocean fronts are small-target entities in SST remote sensing imagery, where the edge information is inconspicuous, variable, and exhibits low contrast, demonstrating weak edge properties.

L408: "depth learning" should be "deep learning"

Thank you very much for pointing out this typographical error. We have thoroughly checked the entire text to ensure consistency in the use of this standard terminology.

L410: The phrase "the recognized front has good independence and integrity" is meaningless.

We have removed this expression.

L414: "small-scale fronts" are mentioned here but scales have not been discussed at all.

The manuscript indeed does not address the scale issue; we have removed this statement.

L420: "The deep learning method outperforms traditional methods in extracting feature parameters such as ocean front intensity and width." – There is no proof in this study. Such claims must be supported by data.

We have revised the manuscript to emphasize that the AI-based approach achieves results comparable to traditional extraction methods, while also addressing the issue of inconsistent threshold selection criteria in gradient-based algorithms.L422:"scale" is mentioned again. See L414.

L422: "scale" is mentioned again. See L414.

The manuscript indeed does not address the scale issue; we have removed this statement.